# Marek's disease virus prolongs survival of primary chicken B-cells by inducing a senescence-like phenotype

**Laëtitia Trapp-Fragnet**[1]*, **Julia Schermuly**[2], **Marina Kohn**[2], **Luca D. Bertzbach**[3,4], **Florian Pfaff**[5], **Caroline Denesvre**[1], **Benedikt B. Kaufer**[3], **Sonja Härtle**[2]*

**1** INRAE, UMR1282 ISP, Equipe Biologie des Virus Aviaires, Nouzilly, France, **2** Ludwig-Maximilians-Universität München, Department of Veterinary Sciences, München, Germany, **3** Institut für Virologie, Freie Universität Berlin, Berlin, Germany, **4** Leibniz Institute for Experimental Virology, Department of Viral Transformation, Hamburg, Germany, **5** Friedrich-Loeffler-Institut, Federal Research Institute for Animal Health, Institute of Diagnostic Virology, Greifswald-Insel Riems, Germany

* laetitia.trapp-fragnet@inrae.fr (LTF); sonja.haertle@lmu.de (SH)

## Abstract

Marek's disease virus (MDV) is an alphaherpesvirus that causes immunosuppression and deadly lymphoma in chickens. Lymphoid organs play a central role in MDV infection in animals. B-cells in the bursa of Fabricius facilitate high levels of MDV replication and contribute to dissemination at early stages of infection. Several studies investigated host responses in bursal tissue of MDV-infected chickens; however, the cellular responses specifically in bursal B-cells has never been investigated. We took advantage of our recently established *in vitro* infection system to decipher the cellular responses of bursal B-cells to infection with a very virulent MDV strain. Here, we demonstrate that MDV infection extends the survival of bursal B-cells in culture. Microarray analyses revealed that most cytokine/cytokine-receptor-, cell cycle- and apoptosis-associated genes are significantly down-regulated in these cells. Further functional assays validated these strong effects of MDV infections on cell cycle progression and thus, B-cell proliferation. In addition, we confirmed that MDV infections protect B-cells from apoptosis and trigger an accumulation of the autophagy marker Lc3-II. Taken together, our data indicate that MDV-infected bursal B-cells show hallmarks of a senescence-like phenotype, leading to a prolonged B-cell survival. This study provides an in-depth analysis of bursal B-cell responses to MDV infection and important insights into how the virus extends the survival of these cells.

## Author summary

Upon MDV entry via the respiratory tract, B-cells are among the first cells to be infected in the lung and allow an efficient amplification of the virus. B-cells ensure the transmission of the virus to activated T-cells in which it replicates and ultimately transforms CD4-positive T-cells. Although playing a pivotal role in the MDV life cycle, the response of B-cells to MDV is currently not fully understood. Here, by using an *in vitro* infection model of primary bursal B-cells, we show that MDV infection leads to a prolonged B-cell survival

**Data Availability Statement:** All relevant data are within the manuscript and its Supporting Information files. Array data are submitted to Array

Express (https://www.ebi.ac.uk/arrayexpress/experiments/E-MTAB-10702).

**Funding:** This research was funded by the Agence Nationale de la Recherche as part of the Animal Health and Welfare ERA-Net MADISUP project awarded to CD (ANR-13-ANWA-0002-01; https://www.anihwa.eu) and the Volkswagen Foundation Lichtenberg grant A112662 awarded to BBK (https://www.volkswagenstiftung.de/en/funding/our-funding-portfolio-at-a-glance/lichtenberg-endowed-professorships). The funders had no role in study design, data collection and analysis, decision to publish, or preparation of the manuscript.

**Competing interests:** The authors have declared that no competing interests exist.

resulting from decreased cell proliferation, protection from apoptosis and activation of autophagy. Our study provides new insights into the B-cell response to MDV infection, demonstrating that MDV triggers a senescence-like phenotype in B-cells that could potentiate their role in MDV pathogenesis.

## Introduction

Marek's disease is a highly contagious disease that affects *galliformes*. In chickens, Marek's disease is associated with a rapid onset of T-cell lymphoma within 3 to 4 weeks post-infection, immunosuppression, and neurological disorders [1–3]. The causative disease agent is the Gallid alphaherpesvirus 2 (GaHV-2) that is widely known as Marek's disease virus (MDV). MDV is an alphaherpesvirus that belongs to the genus *Mardivirus*. Chickens are infected by inhalation of MDV-contaminated dust from the environment [4]. Although the infection in the lungs of chickens remains poorly understood, it is presumed that the virus infects B-cells and antigen-presenting cells (macrophages and/or dendritic cells) that transfer the virus to primary lymphoid organs (bursa of Fabricius, spleen and thymus) [5–8]. B-cells are the primary target cells for cytolytic infection and transmit the virus to activated T-cells [9]. About 10 days post-infection (dpi), the virus establishes a latent phase primarily in CD4+ T-cells [10]. Latency is associated to viral genome integration into the telomeres of the host cells, ensuring lifelong persistence of the virus in the host [11–13]. In addition, MDV can transform latently infected CD4+ T-cells, resulting in T-cell lymphomas that are observed as early as three to four weeks post infection [6,7]. Virus reactivation from latently infected cells can occur at a later stage of infection and is accompanied by a second viremia and continuous shedding of infectious virus from the feather follicle epithelium [7].

During the early cytolytic phase, the virus causes a transient atrophy of the bursa and thymus, which is associated with immunosuppression and lymphopenia in infected birds [14]. This atrophy is characterized by profound histological changes including substantial thinning of the thymic cortex and a degeneration of bursa follicles [14–16]. In addition, this atrophy is tightly associated with a massive depletion of B- and T-cells in these organs which is likely induced by activation of apoptosis [15]. We recently showed that MDV-induced apoptosis in the thymus is mainly triggered in infected cells, while apoptosis in the bursa mostly affects non-infected cells [14]. Moreover, our recent data reveal a significant decrease of cell proliferation in the bursa and a drastic B-lymphopenia in the blood of MDV-infected birds within the first two weeks of infection [14].

B-cells were thought to play a prominent role in MDV life cycle for a long time. We recently demonstrated that mature and peripheral B-cells are dispensable for MDV lymphomagenesis in B-cell knockout chickens [17]. These chickens lack peripheral B-cells but still harbor B-lineage precursors in their bursas and strikingly, we observed that MDV was still able to replicate in the bursa with a delayed virus spread to other lymphoid organs [17,18]. Those findings suggest that bursal B-cells contribute to the establishment of MDV infections and possibly also to an immune response against MDV. Several studies demonstrated that complete removal of B-cells by surgical bursectomy strongly influences MDV viremia and MDV-induced pathogenesis and reduced the vaccine-mediated immune protection [8,19,20]. To date, various studies have reported on the host response to MDV infection in the bursa [21–25]. However, these data were mainly generated from whole bursa tissue samples (containing not only B-cells but also T-cells, macrophages, epithelial cells. . .) collected from infected animals, and therefore the specific response of bursal B-cells to MDV infection remained elusive. We recently

established an *in vitro* infection system for primary bursal B-cells as a powerful tool to decipher how these cells respond to infections with MDV and other viruses [26–28]. We demonstrated that bursal B-cells are permissive to MDV infection in previous studies, with efficient viral replication and high expression of most MDV genes as early as 16 hours post infection (hpi) [27,29].

Herein, we used this *in vitro* system to specifically characterize the bursal B-cell response to MDV infection. Unexpectedly, we discovered that MDV infection extends the lifespan of bursal B-cells in culture. Based on microarray data and functional assays, we demonstrate that this prolonged survival of MDV-infected bursal B-cells is associated with a drastic delay in proliferation/cell cycle progression, a protection from apoptosis and an induction of autophagy-like process. Overall, our data strongly suggest that MDV infection causes B-cells to enter a senescence-like state, which could enable B-cells to survive longer and thereby contributing to the dissemination in MDV in infected chickens. Taken together, our study provides new insights into MDV pathogenesis by unravelling the role of bursal B-cells in the MDV life cycle.

## Results

### MDV infection increases the number of viable B-cells

To determine whether B-cell viability is altered upon MDV infection, primary bursal B-cells were isolated and co-cultured for 24 hours (h) in the presence of soluble CD40L with chicken embryo cells (CEC) infected with RB1B-GFP. Two non-infected control samples were used: (i) uninfected bursal B-cells that we maintained in CD40L-supplemented media in the absence of infected CEC and (ii) the GFP-negative B-cells from infected cultures. At 24 hpi, control (uninfected and GFP-negative) and infected (GFP-positive) B-cells were sort-purified by FACS. Identical numbers of purified cells ($5\times10^5$) were then cultured without CD40L for additional 24 or 48 h. When the number of viable cells was quantified at 48 and 72 hpi (i.e. 24 and 48h post sort purification; Fig 1), we observed in both controls a rapid decrease in the number

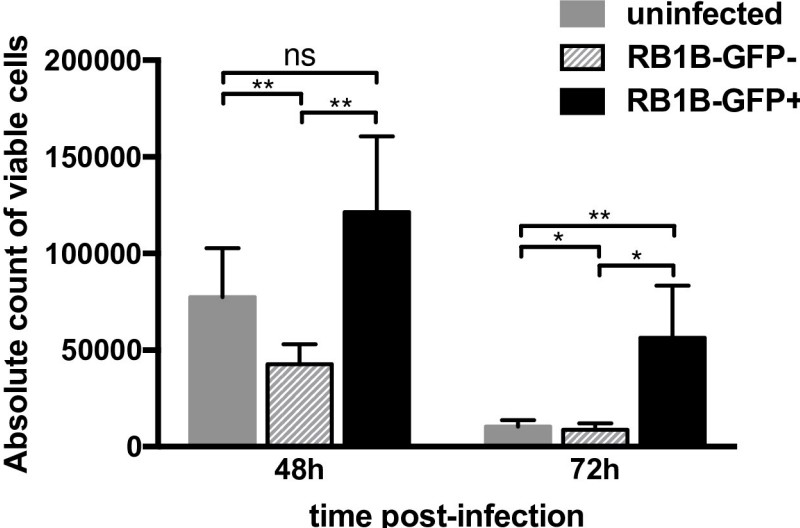

**Fig 1. MDV infection increases the viability of B-cells.** B-cells isolated from bursa were either cultured uninfected or infected with RB1B-GFP. Viable GFP-positive, GFP-negative and uninfected cells were sorted by FACS at 24 hpi and incubated in complete medium without CD40L until further analysis. At 48 and 72 hpi, viable cells were quantified by FACS using BD Trucount tubes. Results obtained from three independent experiments are shown as mean (+/- SD). * p<0.05; ** p<0.005; ns: not significant.

of viable cells from 48 to 72 hpi as expected. In contrast, more cells remained viable in the infected cell population when compared to the non-infected controls. At 48 hpi, we detected $1.2x10^5$ viable cells in the infected population compared to the uninfected ($7.7x10^4$ cells) and GFP-negative cells ($4.3x10^4$ cells). This difference widened further at 72 hpi where we detected a statistically significant 5.4 and 6.5-fold increase of viable B-cells in the MDV-infected population compared to control non-infected cells and GFP-negative cells, respectively.

## MDV infection strongly regulates gene expression in B-cells

The higher number of viable cells observed in the infected population suggested that MDV is either able to prolong primary B-cell survival *in vitro* (independently of CD40L stimulation) and/or to increase B-cell proliferation. To address this question, we performed a comparative microarray analysis on viable MDV-infected versus non-infected B-cells that where sort-purified at 24 hpi. A total of 2,186 probe IDs were found to be differentially expressed in MDV-infected B-cells compared to control B-cells, with 816 up-regulated and 1,370 down-regulated genes in arrays of infected cells (S1 Table). Analysis of the involved signaling pathways revealed that cellular, environmental information, genetic, metabolic processes and organismal systems are affected upon MDV infection in B-cells (Tables 1 and S1). The highest number of differentially expressed genes in MDV-infected B-cells could be assigned to cytokine-cytokine receptor interaction signaling (50 genes) and cell growth and death (171 genes). Interestingly, genes involved in cell cycle (44 genes), cellular senescence (43 genes), apoptosis (32 genes), necroptosis (31 genes) and p53 signaling pathway (21 genes) were mostly down-regulated.

**Table 1. Top 20 canonical pathways of differentially expressed genes in MDV-infected B-cells.**

| KEGG pathway | #genes[a] | Processes | Function | P-value |
|---|---|---|---|---|
| Cytokine-cytokine receptor interaction[b] | 50 | Environmental Information | Signaling molecule and interaction | 4.00E-05 |
| Cell cycle[b] | 44 | Cellular Processes | Cell growth and death | 6.60E-09 |
| MAPK signaling pathway | 44 | Environmental Information | Signal transduction | 6,45E-02 |
| Cellular senescence[b] | 43 | Cellular Processes | Cell growth and death | 2,80E-06 |
| Endocytosis | 42 | Cellular Processes | Transport and catabolism | 2,52E-02 |
| Apoptosis[b] | 32 | Cellular Processes | Cell growth and death | 7,00E-04 |
| Necroptosis[b] | 31 | Cellular Processes | Cell growth and death | 2,10E-03 |
| FoxO signaling pathway | 31 | Environmental Information | Signal transduction | 1,00E-03 |
| NOD-like receptor signaling pathway | 30 | Organismal Systems | Immune system | 3,10E-03 |
| Regulation of actin cytoskeleton | 29 | Cellular Processes | Cell motility | 3,36E-01 |
| Focal adhesion | 29 | Cellular Processes | Cellular community–eukaryotes | 3,29E-01 |
| Lysosome | 27 | Cellular processes | Transport and catabolism | 7,40E-03 |
| Protein processing in ER | 26 | Genetic Information | Folding, sorting and degradation | 1,53E-01 |
| Purine metabolism | 26 | Metabolism | Nucleotide metabolism | 2,74E-02 |
| mTOR signaling pathway | 24 | Environmental Information | Signal transduction | 1,72E-01 |
| Toll-like receptor signaling pathway | 24 | Organismal Systems | Immune system | 9,00E-04 |
| Phagosome | 23 | Cellular Processes | Transport and catabolism | 2,05E-01 |
| Wnt signaling pathway | 23 | Environmental Information | Signal transduction | 2,60E-01 |
| Tight junction | 22 | Cellular Processes | Cellular community–eukaryotes | 5,51E-01 |
| p53 signaling pathway[b] | 21 | Cellular Processes | Cell growth and death | 5,00E-04 |

[a] number of genes differentially expressed assigned to the pathway

[b] pathways involved in cell growth and death are highlighted in yellow

Hence, MDV infection triggers massive gene regulation changes in B-cells that could have an impact on B-cell survival at different levels.

## MDV infection down-regulates expression of cytokine/cytokine-receptor genes involved in proliferation and survival mediation

Survival or proliferation of immune cells is very often regulated by cytokines. Among the differentially expressed genes (DEGs) in infected B-cells that were assigned to the cytokine-cytokine receptor interaction signaling pathway, which contains both cytokines and chemokines, 41 genes were down-regulated and 9 genes were up-regulated (Fig 2 and S2 Table).

Hereof, IL6 that can be produced by activated B-cells [30] was 3-fold up-regulated in infected B-cells (Fig 2A). Our transcriptomic analysis also showed that BAFF (B-cell activating

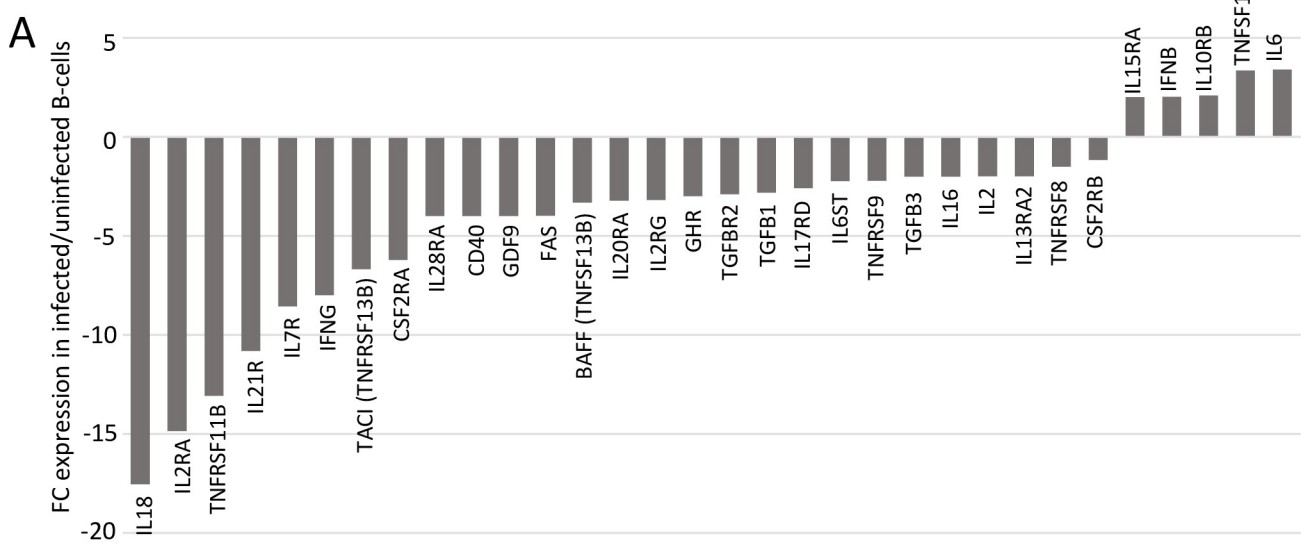

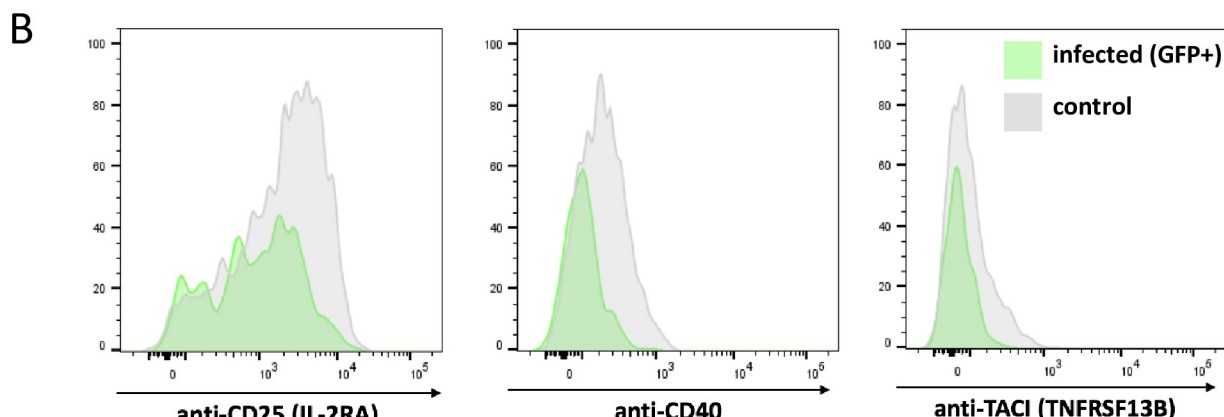

**Fig 2. Cytokine/cytokine receptors are differentially expressed in MDV-infected B-cells.** (A) Microarray analysis revealed that 32 genes involved in cytokine/cytokine receptors interaction pathways are differentially expressed in RB1B-GFP-infected B-cells at 24 hpi. Results are presented as fold change for the mRNA expression of each gene in MDV-infected B-cells relative to uninfected B-cells. (B) Protein expression corresponds to mRNA abundance. Bursal cells were infected with RB1B-GFP. At 24 hpi, cells were stained for surface expression of CD25 (IL2RA), CD40 or TACI and analyzed by flow cytometry. Cells from infected cultures were gated for GFP$^+$ viable, single lymphocytes, uninfected control cultures were gated for viable, single cells. The picture is representative for one out of four independent experiments.

**Table 2. Cytokines showing a proliferative/differentiation activity on avian B-cells.**

| Cytokine | Function | FC | Receptor (s) | FC |
|---|---|---|---|---|
| TNFSF13B (BAFF) | Essential for B-cell survival | -3 | **TNFRSF13B (TACI)**<br>**TNFRSF13C (BAFF-R)** | -7<br>+1 |
| CD40L | Induces B-cell proliferation | ND | **CD40** | -4 |
| IL2 | Induces B-cell proliferation | -2 | **IL2R** consisting of IL2Rα,−β and−γ subunit<br>**IL2RA** expression generates the high affinity IL2R | -15 |
| IL21 | Synergistically enhances CD40L-induced B-cell proliferation | ND | **IL21R** (heterodimer of IL21R and common γ-chain) | -11 |
| IL10 | Synergistically enhances CD40L-induced B-cell proliferation and supports Ig class switch | ND | **IL10R**, heterodimeric receptor composed of **IL10RA** subunit (ligand-binding subunit) and<br>**IL10RB** subunit (accessory) | ND<br>2 |
| IL6 | Supports differentiation to plasma cells | +3 | **IL6-R** (consisting of a complex of IL6-R and the signaling molecule IL6ST/gp130) | ND<br>-2 |

factor of the TNF family, TNFSF13B) and one of its receptors TACI were down-regulated (3- and 7-fold respectively) in infected B-cells while no differential expression was observed for BAFF-R (TNFRSF13C). BAFF is known to promote the survival of splenic and bursal chicken B-cells in culture [31,32] and is produced by B-cells in an autocrine manner. The decrease in BAFF expression therefore suggests that the prolonged survival of B-cells induced by MDV infection is unlikely mediated by BAFF. Similarly, cytokines known to trigger B-cell proliferation (CD40L, IL21, IL2, IL10) and more importantly, proliferation mediating receptors (CD40, IL2RA (CD25), IL21RA) were down-regulated in MDV-infected B-cells (Fig 2 and Table 2).

We could also demonstrate down-regulation of IL2RA, CD40 and TACI expression at the protein level by flow cytometry (Fig 2B). Based on these observations, it seems unlikely that cytokines/cytokine receptors are responsible for the increased viability of infected B-cells.

## The expression of major cell cycle regulators is strongly down-regulated in MDV-infected B-cells

In order to determine whether the higher number of B-cells detected during MDV infection could be attributed to an increase in cellular proliferation, we examined the 44 differentially expressed cell cycle-related genes (Fig 3A and S3 Table) from which 38 genes were down- and only 6 genes were up-regulated. Strikingly, all down-regulated genes are related to cell cycle progression and especially the expression of several cyclins (A1, E2, D1, D3, A2, B2, B3) and the proto-oncogene myc (FC -9) is strongly decreased in MDV-infected B-cells. In contrast, cyclin-dependent kinase inhibitors (p27$^{Kip1}$, p21$^{Cip1}$ and CDKN2B, the avian equivalent to mammalian p16$^{INK4a}$) as well as growth arrest and DNA damage inducible factors (GADD45A and GADD45B) are up-regulated in infected cells. Hence, these data contradict an increased proliferation but rather strongly suggest that proliferation of MDV-infected B-cells is severely impaired.

## MDV delays cell cycle progression in G0/G1

Although we detected an increased number of viable cells in the infected population, microarray data argue for an MDV-induced defect of B-cell proliferation. To confirm that MDV affects the proliferation of B-cells, we quantified BrdU incorporation in purified RB1B-GFP-infected B-cells (Fig 3B). We observed a significant decrease of proliferation of infected B-cells (about 27%) compared to non-infected cells at 48 hpi, and still 18% less proliferation at 72 hpi, a time point where proliferation in cultures without CD40L is already very low.

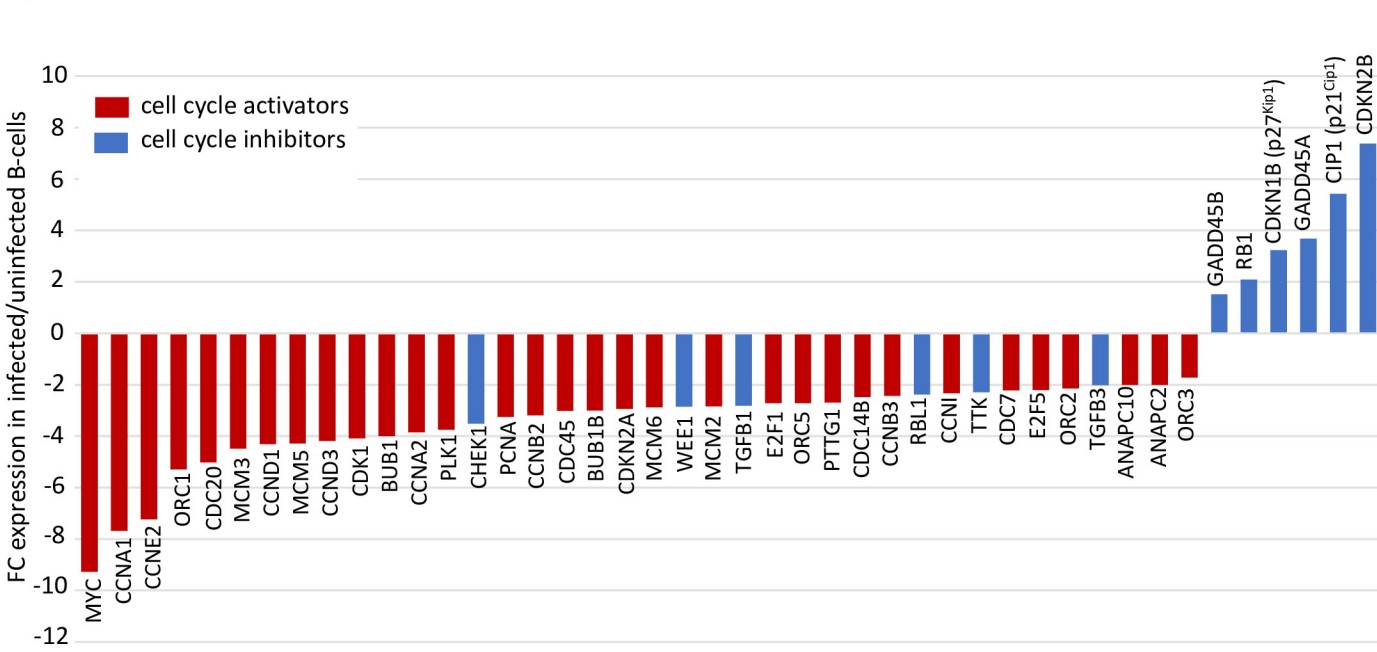

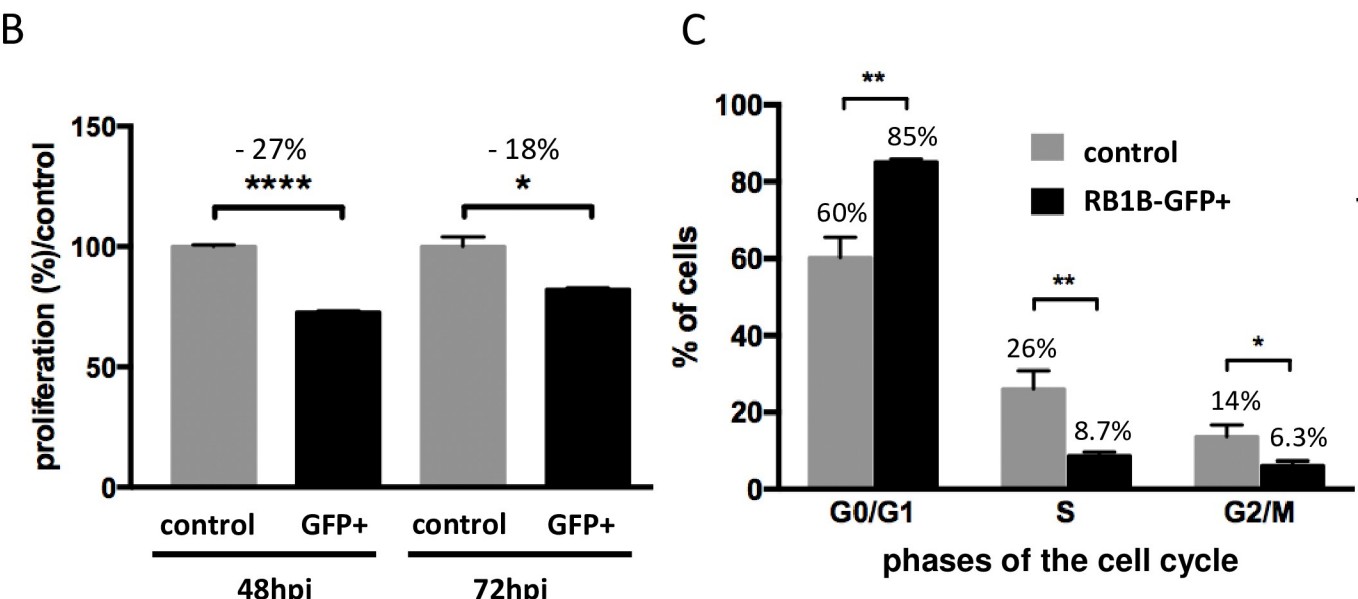

**Fig 3. The cell cycle is severely impaired in MDV-infected B-cells.** (A) Microarray analysis revealed that 44 genes involved in cell cycle regulation pathways are differentially expressed in RB1B-GFP-infected B-cells at 24 hpi. Results are presented as fold change (FC) for the mRNA expression of each gene in MDV-infected B-cells relative to uninfected B-cells. (B-C) MDV infection delays B-cell proliferation and cell cycle progression. Bursal B-cells were infected with RB1B-GFP. At 24 hpi, non-infected cells (control) and infected GFP+ cells were sorted by FACS and maintained in complete media for further analysis. (B) At 32 and 56 hpi, BrdU reagent was added in the control and infected cultures for additional 16h. Cell proliferation was then measured at 48 and 72 hpi by BrdU assay. Results of three independent experiments are presented as the percentage of proliferation (+/- SD) relative to the proliferation of non-infected control cells (set at 100%). (C) At 48 hpi, DNA content was analyzed by cytometry in RB1B-GFP+ and non-infected B-cells. Results corresponding to the percentage of cells in G0/G1, S and G2/M phases of three independent experiments are represented as mean (+/- SD). * $p < 0.05$; ** $p < 0.005$; **** $p < 0.0001$.

To further characterize the decrease in proliferation in infected B-cells, we assessed their progression in the cell cycle (Fig 3C). At 48 hpi, viable infected (GFP-positive) and non-infected B-cells were sorted by FACS and DNA content was analyzed by flow cytometry. Only 8.7% of the infected B-cells were in the proliferative phase of the cell cycle (S-phase), compared to 26% in the non-infected population. Likewise, the population of infected cells in G2/M (6.3%) was lower than in uninfected cultures (14%). Accordingly, more than 85% of infected B-cells remained in cell cycle phase G0/G1 compared to 60% of non-infected cells, demonstrating a significant shift in cell cycle phases in MDV-infected B-cells.

These results confirm our microarray analysis and demonstrate a strong impairment of B-cell proliferation upon MDV infection. Moreover, we could more precisely define that MDV triggers a G0/G1 cell cycle arrest in B-cells, which is in accordance to the reduced expression of cyclins D and E and the increased abundance of cdk inhibitors detected by microarray. Also, this confirms that the increased number of viable B-cells in MDV-infected cultures does not result from an increased cell proliferation rate.

## MDV induces differential expression of apoptosis-related genes

In culture, bursal B-cells show a very high apoptosis rate [33]. The significantly reduced proliferation of MDV-infected B-cells and the associated cell cycle arrest raised the question whether apoptosis of infected B-cells was influenced. Microarray data revealed 32 DEGs involved in apoptosis, among which 26 were down-regulated and 6 were up-regulated (Fig 4A and S4 Table). However, as pro- and anti-apoptotic factors were both up- and down-regulated, transcriptomic data were inconclusive.

To clarify whether or not MDV induces apoptosis in B-cells *in vitro*, sort-purified RB1B-GFP-infected and control B-cells from uninfected cultures were stained with Annexin V/eFluor 780 and analyzed by flow cytometry 48 hpi (Fig 4B). Strikingly, the overall percentage of apoptotic cells in the infected B-cell population was significantly lower (51.3%) compared to the massive apoptosis rate in uninfected cells (96%). About half of the apoptotic infected cells (27% of total) were in an early stage of apoptosis (Annexin V +/eFluor -), while the other half (24% of total) were late apoptotic cells (Annexin V +/eFluor +). In contrast, the vast majority of control cells were late apoptotic cells (95% of total) with only 1% of cells in early apoptosis (Fig 4B). These data demonstrate that MDV infection significantly reduces innately occurring apoptosis in B-cells, leading to prolonged survival of infected cells.

## Meq is dispensable for B-cell survival

MDV oncoprotein Meq was shown to exhibit anti-apoptotic activities and thus might be involved in reduced apoptosis caused by MDV infection. To challenge this hypothesis, we first addressed the question if Meq is expressed in B-cells upon MDV lytic infection. Therefore, B-cells were infected with RB-1B_UL47-RFP-Meq-GFP and expression of UL47 and Meq were detected by flow cytometry at 48 hpi based on RFP and GFP expression, respectively. We observed that most of the lytically infected (UL47-RFP+) cells also expressed Meq-GFP (Fig 5A).

To determine whether Meq could contribute to B-cell survival upon MDV infection, cells were infected with a recombinant MDV deleted for the *meq* gene (RB1BΔMeq-GFP) and compared to B-cells infected with the parental RB1B-GFP virus encoding Meq and non-infected control B-cells. At 24 hpi, viable infected and non-infected control cells were sorted by FACS and cultured for further analysis. When the number of viable cells was assessed by flow cytometry at 48 and 72 hpi, we confirmed that cultures with RB1B-GFP-infected B-cells contained significantly higher number of viable cells compared to uninfected controls (Fig 5B).

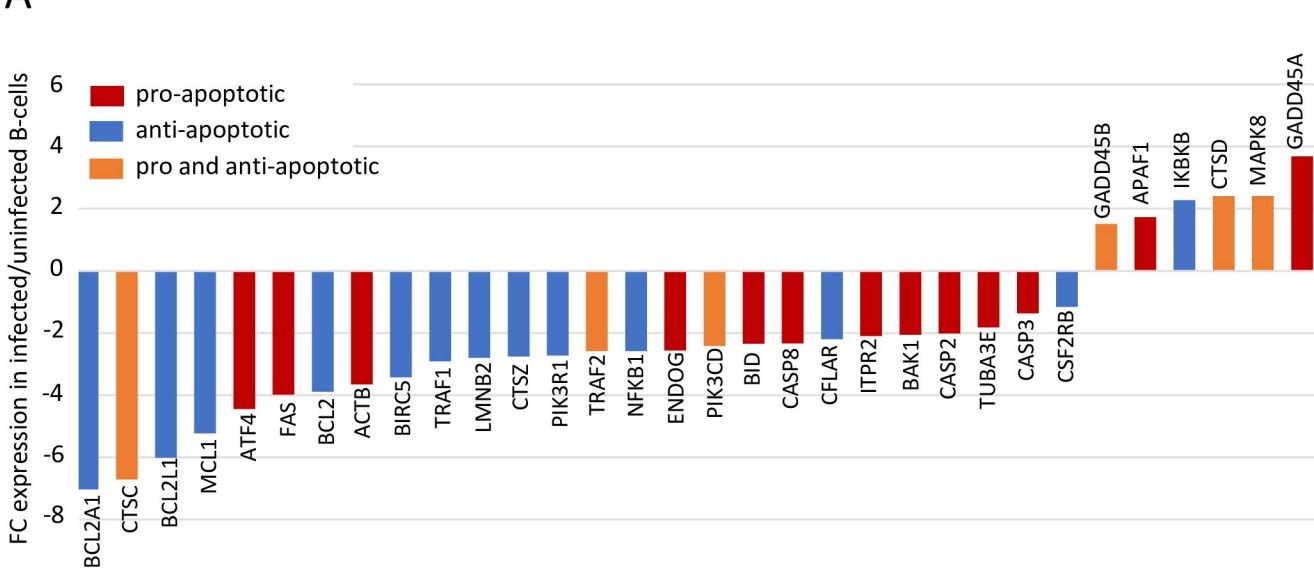

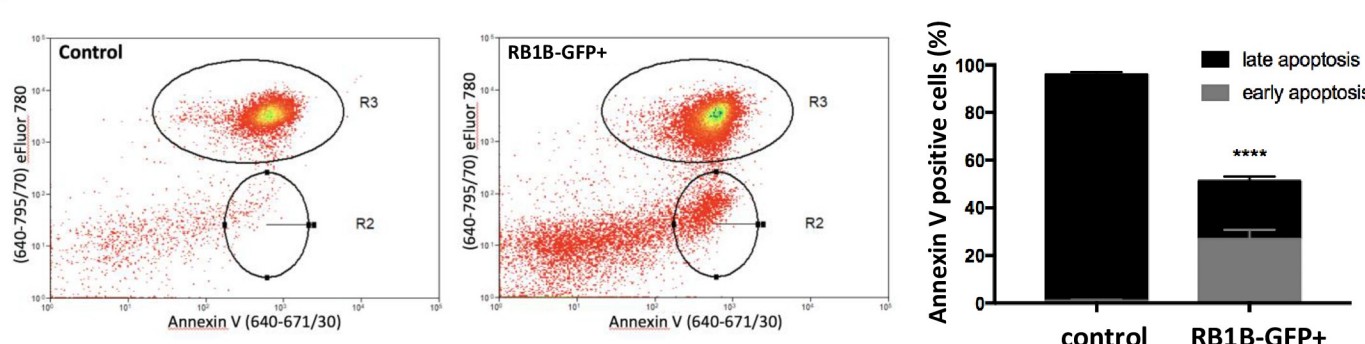

**Fig 4. MDV infection protects B-cells from apoptosis.** (A) Microarray analysis revealed that 32 genes involved in apoptosis are differentially expressed in RB1B-GFP-infected B-cells. Results are presented as fold change (FC) for the mRNA expression of each gene in MDV-infected B-cells relative to uninfected B-cells. (B-C) Bursal B-cells were infected with RB1B-GFP. Viable control and infected (GFP+) cells were sorted by FACS at 24 hpi and incubated for 24 additional hours in complete media. (B) At 48 hpi, cells were stained using the viability dye eFluor 780 and Annexin V to analyze apoptosis by flow cytometry. Representative dot plot histograms obtained from control and infected GFP+ B-cells are shown (left panel) with R2 and R3-gated regions corresponding to early (Annexin+/eFluor-) and late (Annexin+/eFluor+) apoptotic cells, respectively. Means (+/- SD) from three independent experiments are presented as stacked bars (right panel). **** p<0.0001.

Interestingly, we found that the infection with RB1BΔMeq-GFP did not result in a significant difference in the number of viable cells compared to RB1B-GFP (encoding Meq) infection. This demonstrates that Meq does not contribute to prolonged B-cell survival.

## MDV infection induces autophagy in B-cells

As we could show that reduced apoptosis in MDV-infected B-cells is not mediated by Meq, we were looking for alternative explanations. Autophagy is involved in degradation and recycling of intracellular components and therefore is considered as a potent pro-survival mechanism. In addition, the autophagy and apoptosis pathways have been shown to be tightly interconnected, and under many physiological conditions, autophagy antagonizes apoptosis [34,35]. Further analysis of our transcriptome data revealed 21 DEGs involved in the autophagic pathway

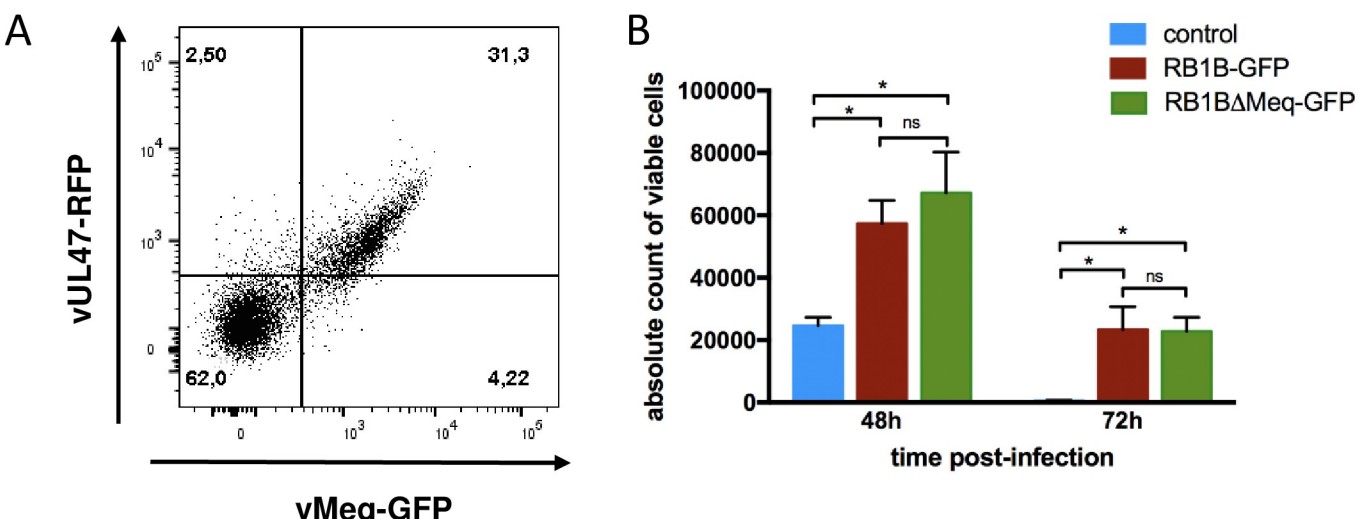

**Fig 5. Meq is dispensable for MDV-induced B-cell survival.** (A) Meq expression was assayed by flow cytometry at 48 hpi in B-cells infected with RB-1B_UL47-RFP-Meq-GFP. The samples were gated for viable, single, AV20+ cells. (B) Viable B-cells infected with the parental RB1B-GFP or RB1BΔMeq-GFP and non-infected cells were sort purified by FACS at 24 hpi. Cells were cultured for additional 24, 48 and 72 h. Viable cells were quantified by cytometry using BD Trucount tubes at 48 and 72 hpi. Results obtained from two independent experiments are shown as mean (+/- SD). * p<0.05; ns: not significant.

in MDV-infected B-cells (Fig 6A and S5 Table). The 9 down-regulated genes are mainly involved in inhibition of autophagy (e.g. BCL2 family members, PIK3). In contrast, the 12 up-regulated genes are described as activators of autophagy such as DEPCD6, an mTOR inhibitor which was 16-fold up-regulated in infected B-cells. To assess a possible induction of autophagy in B-cells in response to MDV infection, viable non-infected and RB1B-GFP-infected B-cells were purified by FACS at 24 hpi, centrifuged at low speed on coverslips and immunostained with the Lc3B autophagosomal marker (Fig 6B). To validate the detection of Lc3-II, uninfected B-cells were treated with rapamycin, a potent inducer of autophagy followed by treatment with bafilomycin A1 to block the autophagic flux. In this control, we could readily observe that Lc3-II accumulated as cytoplasmic puncta in treated cells (Fig 6B). In about 60% of the infected B-cells, we also detected a strong punctuated cytoplasmic Lc3-II staining indicating an accumulation of autophagosomes, which we did not observe in the non-infected control cells (Fig 6B and 6C). These data thus confirmed that autophagy is induced in B-cells after MDV infection.

## MDV triggers a senescence-like phenotype in infected B-cells

MDV-infected B-cells exhibit a higher survival rate associated with reduced proliferation mediated by cell cycle withdrawal in G0/G1, activation of autophagy and protection against natural apoptosis. Strikingly, this particular phenotype of infected B-cells is very similar to descriptions of cellular senescence. This cellular process plays a key role in regulating cellular lifespan and is defined as a non-proliferative but viable state of the cell [36,37]. Indeed, micro-array analysis revealed that 43 DEGs in MDV-infected B-cells were assigned to the senescence pathway (Fig 7 and S6 Table). Twenty of these genes are cell cycle regulators, contributing to the above described cell cycle arrest, which is a hallmark of senescent cells. Although functionality of senescence-related genes is not yet fully elucidated, we identified several anti-senescence factors (e.g. ZFPL36L1, NFATC1, MYBL2, NBN, FOXM1, SLC25A5 and FOXO1) that were down-regulated (up to 4-fold) in MDV-infected B-cells.

A special feature of senescent cells is the senescent associated secretory phenotype (SASP), where cells secrete high levels of inflammatory cytokines (especially IL6 and IL8), immune

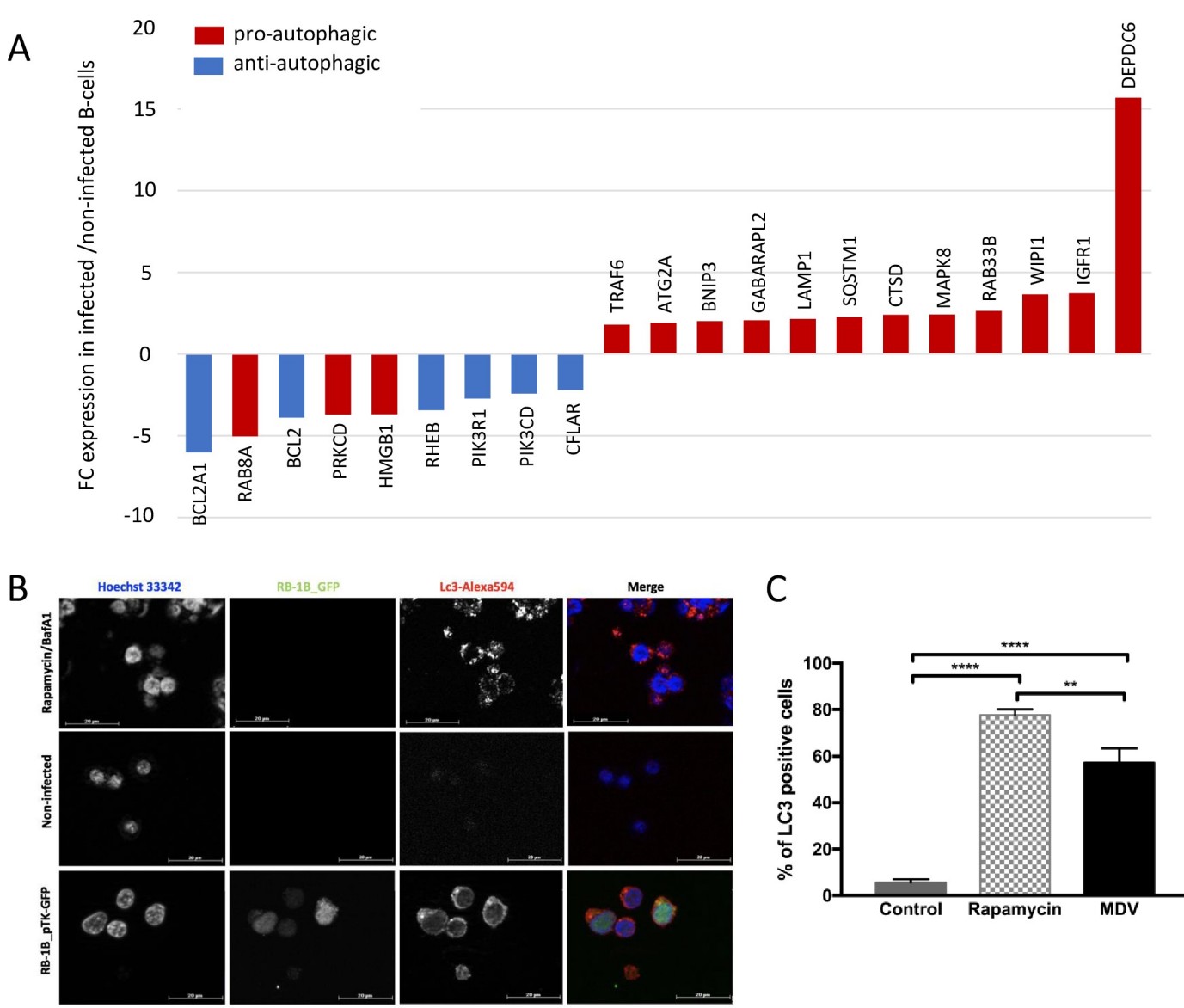

**Fig 6. Autophagy is induced in MDV-infected B-cells.** (A) Microarray analysis revealed that 21 genes involved in autophagy are differentially expressed in RB1B-GFP-infected B-cells. Results are presented as fold change (FC) for the mRNA expression of each gene in MDV-infected B-cells relative to uninfected B-cells. (B) Expression and localization of the autophagosomal marker Lc3 in B-cells infected with RB1B-GFP. At 24 hpi, viable control and infected cells were sorted by FACS prior to be subjected to immunofluorescence using a rabbit anti-Lc3 monoclonal antibody and an Alexa Fluor 594-conjugated secondary antibody (red). Non-infected B-cells treated with rapamycin/bafilomycin A1 (BafA1) were used as positive control for autophagy induction. Nuclei were stained with Hoechst 33342 (blue), and infected cells expressing GFP were directly visualized by fluorescence microscopy (green). (C) Proportion of cells in autophagy. Results are shown as percentage of Lc3-positive cells (+/- SD). ** $p < 0.005$; **** $p < 0.0001$.

modulators, growth factors and matrix metalloproteinases [36–38]. Strikingly, aside the increased in IL6 mRNA expression in infected B-cells (Fig 2A), we also found that IL8 (CXCL8; IL8L2, CXCLi2) showed the strongest up-regulation of all DEGs (with an FC of 47) and that the mRNA abundance for K203 (chCCLi3), another chicken inflammatory chemokine was 26-fold increased (S1 Fig). In addition, the expression of matrix metallopeptidases MMP-2, -7 and -9 was up-regulated until 9-fold in MDV-infected B-cells compared to uninfected cells.

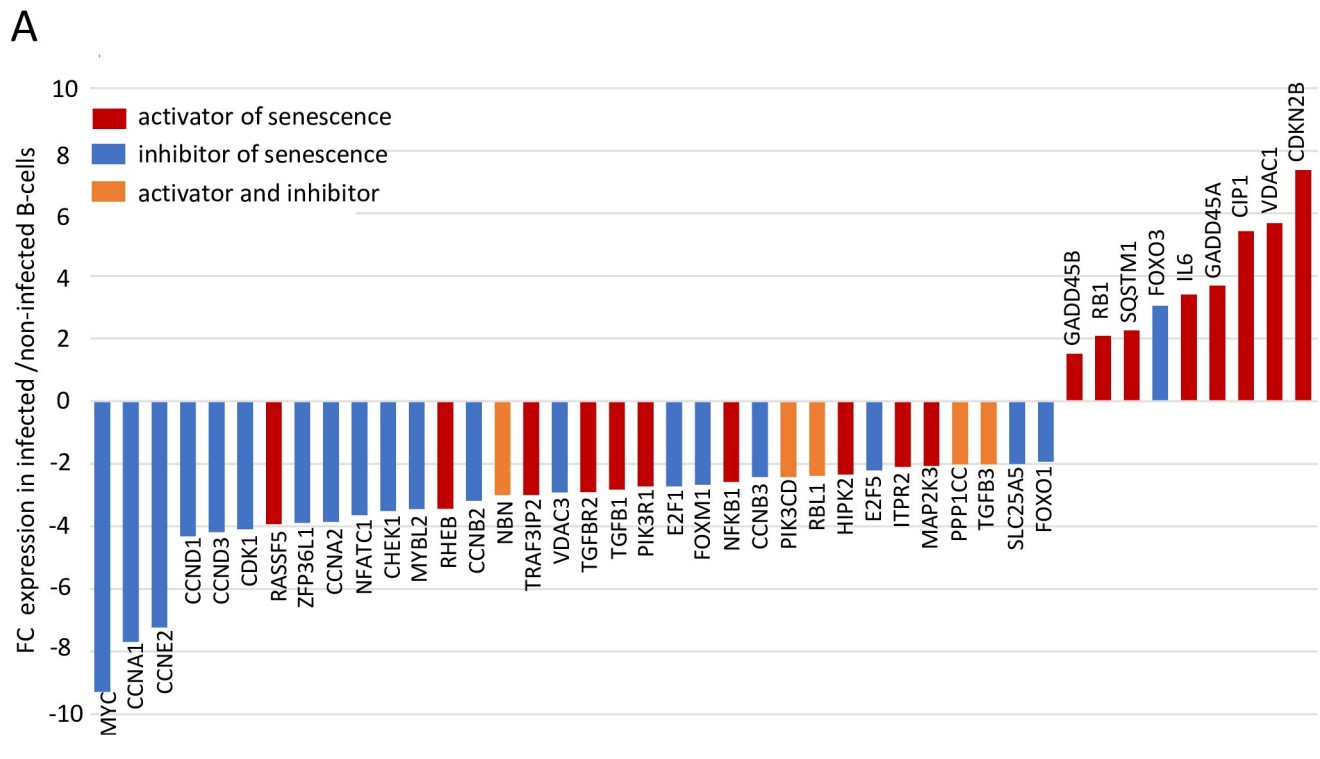

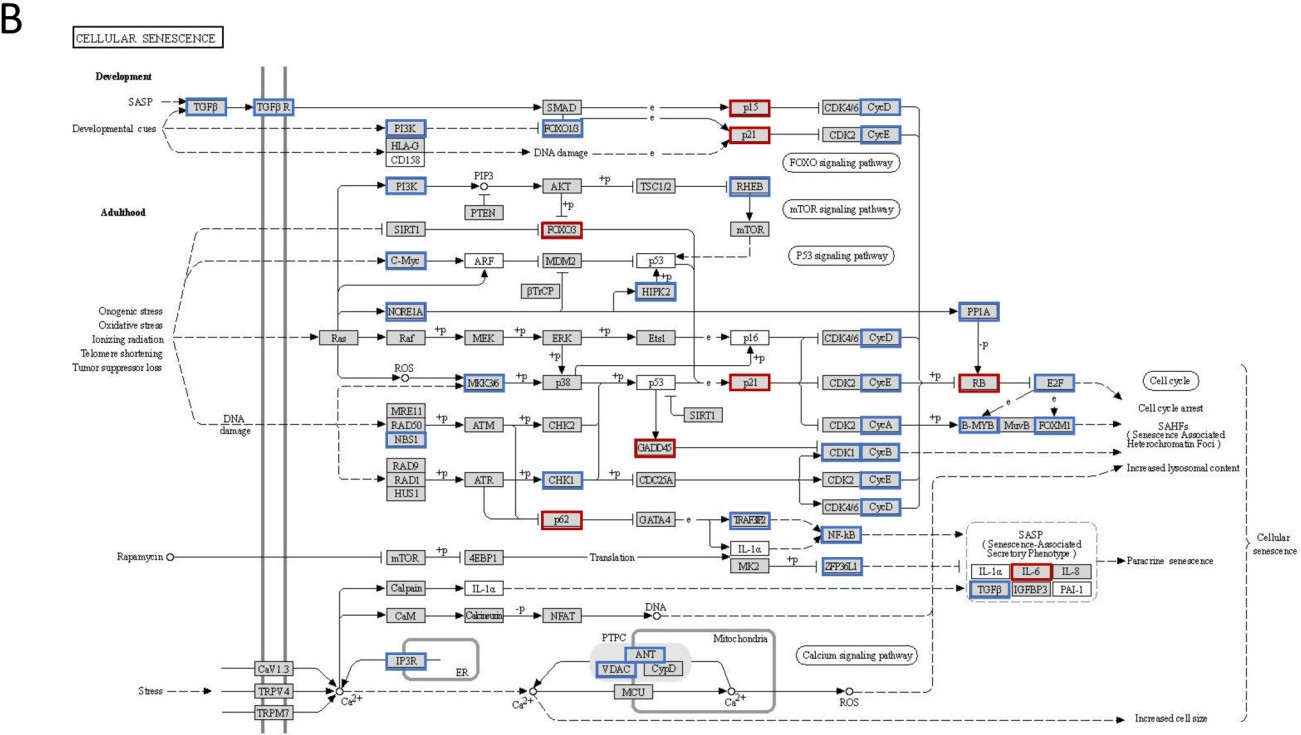

**Fig 7. Cellular senescence related-genes are differentially expressed in MDV-infected B-cells.** (A) Microarray analysis revealed that 43 genes involved in senescence are differentially expressed in RB1B-GFP-infected B-cells. Results are presented as fold change (FC) for the mRNA expression of each gene in MDV-infected B-cells relative to uninfected B-cells. (B) KEGG pathway for cellular senescence. Genes identified in our microarray are highlighted by blue (down-regulated) and red (up-regulated) boxes.

Thus, our transcriptomic analysis strongly suggests that MDV infection drives B-cells into a senescence-like state, providing a comprehensive explanation for their higher survival rate *in vitro*.

## Discussion

It has been previously shown that MDV infection *in vivo* induces an early and transient atrophy of the primary lymphoid organs [14,15,39]. In particular, a significant reduction of bursa follicle size associated with a massive depletion of B–cells in the medulla was detected between 10 and 14 dpi [14]. This destructive effect of MDV infection in the bursa resulted in a reduction of circulating B-cells and an alteration of the antibody response to various antigens [40–42]. But until now, the MDV life cycle and the host response mediated by MDV infection specifically in bursal leukocytes were poorly investigated, mostly due to the lack of suitable *in vitro* models with primary target cells. Here, we took advantage of our recently established B-cell infection model to provide insights into the bursal B-cells response to MDV infection. With this infection model, we have repeatedly demonstrated that bursal B-cells efficiently support infection and lytic replication of MDV *in vitro* [27,29,43,44].

Based on these data and the fact that chicken B-cells in culture die rapidly from apoptosis, it was expected that the observed high MDV infection levels would lead to even more increased B-cell death. Surprisingly, we detected that MDV-infected B-cell cultures contained more viable cells than the controls. Generally, this increase in the number of viable cells could be either caused by increased cell proliferation or reduced cell death. To elucidate the underlying cellular processes, we performed transcriptomic analyses of sort-purified MDV-infected and non-infected B-cells. We found that pathways for cytokine-cytokine receptor interaction and cell cycle were significantly affected. Since the differentiation and maturation of B-cells is largely controlled by cytokines, we assumed that MDV-induced changes in the cell response to the cytokine could cause the increased number of viable cells. However, we did not find up-regulation of pro-survival or proliferation associated cytokines or their receptors. In contrast, BAFF, the major survival factor for bursal B-cells, which is produced by B-cells in an autocrine manner [32], was equally down-regulated as one of its receptors (TACI), while expression of the second receptor BAFF-R remained unchanged. In mammals, infections with many pathogens induce the expression of BAFF and/or its cognate receptors (BAFF-R, TACI and APRIL), thus promoting the survival, proliferation and activation of B-cells and the increased production of pathogen-specific antibody [45] but this mechanism is not activated by MDV. Stimulation of CD40 on the B-cell surface with CD40L (CD154) induces strong proliferation of bursal B-cells [46]. However, both mRNA and protein expression of CD40 were reduced in infected cells, which likely made them less sensitive to stimulation of CD40L. Likewise, the expression of IL2RA and IL21R, two other receptors that could mediate B-cell proliferation were also decreased. Therefore, expression data for cytokines /cytokine receptors make it unlikely that increased viability in MDV-infected cultures is driven by cytokines.

As cell cycle related gene expression was also highly modified, we analyzed cell cycle progression of MDV-infected cells and evidenced a decrease of cell proliferation associated with a delay in cell cycle progression in G0/G1. This is consistent with histological observations of bursas collected from MDV-infected chickens, which showed a global decrease of cell proliferation in the medulla [14]. We and others have previously reported that critical regulators of the cell cycle are tightly controlled upon MDV infection *in vitro* and *in vivo*, depending on the cell type/tissue and the stage of infection [47,48]. In lytically infected CEC *in vitro*, we have demonstrated that MDV induces an S-phase arrest, which could facilitate MDV replication [48]. In contrast, here we found that MDV-infected B-cells, the natural target cells of the virus,

accumulated preferentially in G0/G1, indicating either a cell cycle withdrawal in G0 or an arrest of the cell cycle progression at the G1/S checkpoint. This result was supported by the strong down-regulated expression of the cyclins/cdk involved in cell cycle progression and the up-regulation of p21 and GADD45. Of note, a similar transcriptomic profile of expression of those cell cycle regulators has previously been published for thymus tissues of MDV-infected birds [47], suggesting that MDV could regulate the cell cycle similarly in both B- and T-cells. Herpesviruses typically trigger a cell cycle arrest at the end of the G1 phase or at the G1/S transition, hence blocking the replication of cellular DNA, and allowing the virus to benefit from a favorable environment for viral DNA replication (for review [49]).

*In vivo* data did not only show a reduced proliferation in infected bursas, but also a strong increase of apoptosis in the medulla [14]. A possible explanation for this discrepancy could come from the observation that *in vivo*, apoptotic cells were mainly not infected and predominantly restricted to the vicinity of MDV-infected cells [14]. This suggests that infected cells might influence uninfected bystander cells by paracrine signaling or by contact as it has been described for other viruses [50,51]. In striking contrast to the massive apoptosis observed in the bursa of infected chickens, we found that MDV protects infected B-cells from death. Therefore, combined with cell cycle withdrawal, anti-apoptotic mechanisms could also participate to the increased survival of MDV-infected B-cells.

Due to its anti-apoptotic abilities, we speculated that the viral oncoprotein Meq could be involved. Meq is a key player in cell transformation, but its role during the lytic phase of infection is not yet established [52]. It was also shown to interact with a number of cell cycle regulators, especially with the tumor suppressor protein p53 leading to inhibition of its transcriptional activity and apoptosis [53,54]. The anti-apoptotic activities of Meq are also implemented through the regulation of gene expression directly involved in apoptosis (e.g. Bcl2, Bcl-XL) and/or through activation of the PI3K/Akt signaling pathway [54–56]. Here, we found that the protein Meq is well expressed in lytically infected B-cells *in vitro*, which is consistent with the previously reported detection of Meq mRNA in these cells [29]. However, experiments using a Meq deletion mutant resulted in equal increased B-cell survival as infections with the wild-type virus, demonstrating that MDV pro-survival activity is not mediated by Meq.

Altogether, our data converge on a particular phenotype of B-lymphocytes induced by MDV infection. Our transcriptomic and functional analysis of MDV-infected B-cells identified several hallmarks classically found in senescent cells. Cellular senescence is mainly characterized by a prolonged and irreversible cell-cycle arrest with secretory proficiency, macromolecular damage and altered metabolism [36,37]. We showed that MDV infection induced a G1/S cell cycle arrest in B-cells that lasted at least until 48 hpi. Based on our microarray analysis, this arrest might be mediated by a major down-regulation of cyclins involved in cell cycle progression, and a strong up-regulation of p21$^{Cip1}$ and CDKN2B, two cdk inhibitors which are often found at high level in senescent cells [57]. Moreover, we could observe that MDV-infected B-cells showed a strong mRNA up-regulation of certain interleukins/chemokines (IL6 and IL8) and matrix metallopeptidases. This is consistent with the previously described senescent associated secretory phenotype (SASP) [58]. Another characteristic of senescent cells is the outcome of macromolecular damage such as DNA damage triggered by telomeres shortening or by genotoxic stress. We previously demonstrated that MDV induces double strand breaks in the host genomic DNA during lytic infection *in vitro* and in PBMCs of infected animals during lytic infection [48,59]. In addition, senescent cells accumulate protein damage and establish associated degradation processes such as ubiquitin proteasome system or autophagy [60,61] and we found an accumulation of the autophagosomal marker Lc3-II in MDV-infected B-cells. Furthermore, autophagy inhibitors (e.g. BCL2 family members [62,63]

or RAB8A [64]) were down-regulated, while pro-autophagic factors like DEPDC6 [65,66]; IGF1R [67] or WIPI1 [68] were strongly up-regulated. Thus, we could demonstrate that autophagy is activated in these cells. Often, metabolic pathways are also deregulated in senescent cells. And indeed, pathway analysis indicates that lipids, carbohydrates, amino acids metabolisms and lysosome secretion are affected upon MDV infection (S1 Table). Altogether, our data demonstrate that the pro-survival ability of MDV-infected B-cells actually relies on the establishment of a cellular senescence-like phenotype. Further studies are needed to identify the molecular mechanisms leading to this particular phenotype and to understand how viral infection increases B-cell viability. However, it is tempting to speculate on viral factors that could be involved in this process. For instance, the expression of the tegument protein VP22 in B-cells upon infection might impair cell proliferation by triggering a cell cycle arrest, as we previously observed in CEC [48]. Moreover, it is well known that various cellular pathways are tightly regulated by kinases and phosphatases. We could thus assume that the two MDV-encoded serine-threonine kinases US3 and UL13 might contribute to the survival of B-cells, especially US3 that possesses anti-apoptotic properties [69]. The viral telomerase RNA encoded by MDV might also be a factor that could influence the lifespan of infected B-cells not only due to its telomerase activity but also through a potential anti-apoptotic activity as described for hTR [70]. For a long time, cellular senescence was only associated to telomere attrition. However recent reports have shown that it can indeed be induced by various events leading to genomic instability and DNA damage responses, such as oncogene expression or increased production of reactive oxygen species (ROS) [36]. We previously showed that the cell cycle modulation activity of VP22 is strongly associated to its ability to induce DNA damages in cells [48,59]. We had also demonstrated that MDV infection is associated with increased ROS and nitrogen species levels that might thus also contribute to the establishment of the senescence-like phenotype observed in infected B-cells.

Many herpesviruses (e.g. VZV, HCMV, PRV, EBV or KSHV) were shown to sustain the survival of infected cells either to facilitate their own replication or to persist in cells [71–75]. Interestingly, herpesviruses such as the oncogenic γ-herpesviruses Epstein-Barr and Kaposi sarcoma herpesvirus (KSHV), as well as the cytomegalovirus ß-herpesvirus, are also able to trigger and/or to suppress senescence-like responses depending of the cell type infected and the stage of infection [76–83]. Senescence can act as an anti-viral defense, leading some viruses to develop strategies to subvert or prevent senescence. On the contrary, other viruses such as the alphaherpesvirus VZV, exploit the senescence response to enhance their replication [84]. So far, the impact of lymphotropic herpesvirus on senescence was mostly studied in epithelial cells, endothelial cells or fibroblasts [76–82]. Here we report for the first time the induction of a senescence-like phenotype by a herpesvirus in primary B-cells. Though on first glance a peculiar feature, the senescence-like phenotype of bursal leukocytes could promote MDV persistence of the virus thus allowing it to have more time to recruit and infect T-cells. First, due to the short life of bursal B-cells, which might be too short for MDV replication, one can assume that MDV developed strategies to extend the lifespan of B-cells in order to complete its replication and a successful morphogenesis. In addition, the prolonged survival of B-cells could favor the transmission of the virus from B- to T-cells by hampering cell death until T-cells were recruited to the bursa.

In summary, our work emphasize the importance of studying pathogens in their natural target cells, as the cell responses induced by infection differ between cell types and *ex vivo* analysis of whole tissue can be masked by strong bystander effects. Our study sheds a new light on the specific response of bursal B-cells to MDV infection by demonstrating that lytic MDV infection leads to prolonged survival of B-cells probably through the establishment of a senescence-like state. This particular phenotype might ultimately allow infected B-cells to transmit

the virus during an extended period of time and thus promote viral expansion. A better understanding of the cellular responses to MDV infection in lymphoid organs and cells will provide new insights on the pathogenesis of MDV and open new perspectives to develop more effective vaccines against MDV.

## Materials and methods

### Cells and viruses

*In vitro* infection of B-cells with MDV was performed as previously described [27]. Briefly, lymphocytes were isolated from bursa of Fabricius of 6 to 8 weeks-old M11 (B2/B2) chickens (kindly provided by S. Weigend, Friedrich-Loeffler-Institut, Mariensee, Germany) or SPF white leghorn chicks (B13/B13 haplotype provided by the animal facilities of the PFIE, INRAE, Nouzilly). This procedure was carried out in strict compliance with the European legislation for animal experiments (Directive 2010/63/EU), which states that the use of animals solely to sample organs is not submitted to any ethic regulations in Europe (Article 3). Lymphocytes were obtained by dissociation of the bursa and collecting the cells by density gradient centrifugation as previously described [85]. B-cells ($1\times10^7$/ml) were cultured at 41˚C in B-cell medium (Iscove's modified Dulbecco's medium (IMDM), 100 U/ml penicillin,100 µg/ml streptomycin, 8% (vol/vol) fetal bovine serum (all Bio&Sell, Nuremberg, Germany) and 2% (vol/vol) chicken serum (Thermo Fisher Scientific, Waltham, USA)) and activated using recombinant soluble chCD40L [46]. These non-infected B-cells were used as negative controls in all experiments.

For B-cell infections, we used previously published fluorescent reporter viruses of the very virulent RB-1B MDV strain (RB-1B_UL47-GFP; RB-1B_pTK-GFP and RB-1B_UL47-RFP_-Meq-GFP) and a RB-1B Meq-deletion mutant (RB1BΔMeq-GFP), which was derived from RB-1B-pTK-GFP [27,86]. For the sake of clarity, the RB-1B_UL47-GFP and RB-1B_pTK-GFP recombinant viruses are referred to as RB1B-GFP throughout the paper. All viruses were transfected, propagated and titrated in CEC [87]. B-cells ($1\times10^7$) were co-cultured with $2.5\times10^4$ plaque-forming units (pfu) of MDV-infected CEC in 24-well plates for 24 h in the presence of chCD40L.

### Cell sorting and flow cytometry analysis

For cell sorting, cells from non-infected and infected cultures were harvested at 24 hpi, stained with mouse-anti-chB6 coupled to Alexa-Fluor-647 (clone AV20, Southern Biotech Associates, Birmingham, USA) and the Fixable Viability Dye eFluor 780 (Thermo Fisher Scientific) according to standard procedures. From uninfected control cultures viable B-cells were sort-purified by FACS gating on eFluor 780-negative, chB6-positive, single lymphocytes. From infected cultures, viable non-infected (GFP-negative) and infected (GFP-positive) chB6+ B-cells were isolated. Sorting was performed on a FACS Aria III using the FACSDiva software (Becton Dickinson, Franklin Lakes, USA) to a purity of at least 96%.

To analyze surface expression of differentially expressed genes, cells were stained with antibodies for CD25 (28–4, IgG3; [88]) and TACI (1H4, IgG2b), followed by Alexa Fluor 647-conjugated isotype specific secondary antibodies. To generate the anti-TACI antibody, mice were immunized with HEK293T cells transfected with a chicken TACI-FLAG construct. Murine spleen cells were fused to SP2/0 cells and supernatants of resulting hybridomas were tested by flow cytometry on transfected and untransfected HEK293T cells. Monoclonality was ensured by subcloning using the single-cell limited dilution method. As B-cell cultures were stimulated with recombinant CD40L, the CD40 molecule on B-cells was partially blocked by bound CD40L. Hence, to evaluate CD40 surface expression, we stained the cells with an antibody for CD40 (AV79, IgG2a; Bio-Rad laboratories; Hercules, USA) followed by an isotype specific

Alexa Fluor 647-conjugated secondary antibody. In parallel, CD40L molecules bound to CD40 were detected by staining for the murine CD8-tag of CD40L (rat-anti-muCD8a, Ly2, Biolegend, San Diego, USA) followed by an anti-rat-IgG- Alexa Fluor 647 conjugated secondary antibody. Dead cell exclusion was done with Fixable Viability Dye eFluor 780.

Flow cytometry analyses were performed with a FACSCanto II (Becton Dickinson) or MoFlo Astrios$^{EQ}$ flow cytometer (Beckman Coulter, Brea, USA). Data were analyzed using FACSDiva (Becton Dickinson) and FlowJo (FlowJo LLC, Ashland, USA) softwares. Cell cycle data were analyzed using the MultiCycle AV software (Phoenix Flow Systems, California, USA).

### RNA isolation and microarray analysis

For microarray analysis, five independent cell-sorting experiments were performed from uninfected B-cell cultures and infected CEC/B-cells co-cultures at 24 hpi. Three million viable non-infected B-cells, RB1B-GFP-infected B-cells and GFP-negative B-cells were purified and total RNA isolated using pegGoldTrifast (VWR, Darmstadt, Germany) according to the manufacturer's Trizol protocol. Quantity and purity of extracted RNA was determined with a Nano-Drop 1000 (VWR), while the RNA quality was analysed using a 2100 Bioanalyzer (Agilent, Santa Clara, USA). Only RNA samples with an RNA integrity number (RIN) exceeding 8 were used for microarray analysis.

Microarray analysis was performed using Agilent 8x60K chicken-genome microarrays supplemented with 1,699 selected genes out of the chicken genome, supposed to play a crucial role in the chicken immune system (as described in [89]).

After hybridizing, washing and scanning of the microarrays image following manufacturer's instructions (Agilent), arrays were further processed with Feature Extraction Software 10.5.1.1 (Agilent). Further analyses was done using Bioconductor „R"and „limma"(*Linear Models for Microarray Data*). Genes were considered as differentially expressed (DEG) with an at least 2 fold expression difference and a false discovery rate (FDR) <1% (corresponding to an adjusted p-value < 0.01). Fold changes (FC) were calculated from the limma derived coefficients using the formula FC = 2^[(mean group 2)-(mean group 1)]. For a better presentability, FC values below 1 were converted into negative values according to the formula [-1/FC]. FC were either calculated according to non-infected or GFP-negative B-cells, which were co-cultured with infected CEC. Data obtained from these two controls revealed that more than 81% of all genes were similarly regulated in both conditions (S1 Table). In view of the little difference observed between the two controls, most of the biological experiments were carried out using uninfected B-cells as a negative control.

Array data were submitted to Array Express (https://www.ebi.ac.uk/arrayexpress/experiments/E-MTAB-10702/). Annotation of differentially expressed genes into functional pathways was performed using KEGG mapper (https://www.genome.jp/).

### Lymphocyte survival assay

Sort-purified infected and non-infected B-cells (2x10$^5$ cells in 200 µl/well) were cultured at 40˚C in 96-well plates (Thermo Scientific Scientific) in B-cell medium without chCD40L for 24 and 48 h, corresponding to 48 and 72 hpi respectively. For the sake of clarity, all results and figures will refer to post-infection (and not post-sorting) time points. To determine the number of viable cells at the indicated time points, 150 µl from each well were thoroughly suspended and transferred to Trucount tubes (Becton Dickinson, Heidelberg, Germany), adding fixable Viability Dye eFluor 780 for dead cell exclusion. Cells were analyzed by flow cytometry and the number of viable cells was calculated according to the manufacturer's instructions.

## Cell cycle—DNA content analysis

At 24 hpi, viable B-cells from non-infected cultures (control) and viable RB1B-GFP-infected B-cells were sort-purified and $5x10^5$ cells per 24-well plates were incubated in B-cell medium without chCD40L for 24h. At 48 hpi, infected and non-infected B-cells were fixed in ethanol 70% at 4°C for 24 h and RNAs were discarded by Ribonuclease A (Sigma-Aldrich) treatment at 37°C for 1 h. DNA content was detected by staining with 10 mg/ml propidium iodide (Invitrogen, Waltham, USA) and analyzed by flow cytometry [48].

## Apoptosis/cell viability staining

Viable B-cells (control) from non-infected cultures and infected GFP+ cells were sorted by FACS at 24 hpi and $5x10^5$ cells were seeded per P24-wells in B-cell medium without chCD40L for 24 h. At 48 hpi, cells were washed once in phosphate-buffered saline (PBS) and incubated with the Fixable Viability Dye eFluor 780. Cells were then washed twice in PBS 1X, BSA 0.1%. Apoptotic cells were stained using the eBioscience Annexin V Apoptosis Detection Kit APC according to the manufacturer's protocol (Thermo Fisher Scientific). Cell viability and apoptosis analyses were performed by cytometry. Early apoptotic cells were defined as Annexin V-positive / eFluor-negative and late apoptotic cells were defined as Annexin V-positive/eFluor-positive.

## Cell proliferation–bromodeoxyuridine (BrdU) assay

Cell proliferation was assessed using the ELISA BrdU colorimetric BrdU assay (Roche) according to manufacturer's instructions. Bursal B-cells non-infected or GFP+ infected cells were sorted by FACS at 24 hpi and maintained at a density of $2x10^5$ cells per P96-wells in B-cell medium without chCD40L at 40°C. The BrdU labeling solution (1/1000) was added to the culture medium 8 h or 32 h after plating (corresponding to 32 h or 56 hpi) and the cells were incubated for additional 16 h prior to proceed with the protocol recommended by the manufacturer. The absorbance of the samples was measured in an ELISA-reader (Multiskan Ascent, Thermo Labsystem) at 370 nm. Results obtained from infected cells were normalized to those from non-infected cells and are expressed as means (+/-SD) from three independent experiments each done in triplicates.

## Autophagy monitoring by fluorescence microscopy

Autophagy was determined in viable non-infected and MDV-infected B-cells sorted by FACS at 24 hpi. As positive controls of autophagy induction, non-infected bursal B-cells were cultured for 24 h in the presence of 50 nM of rapamycin, and bafilomycin A1 (100 nM, Sigma-Aldrich) was added 4 h prior to harvesting the cells in order to block the degradation of autophagosomes and to detect their accumulation. Cells were plated by centrifugation using a Cytospin 4 (Thermo Fisher Scientific) for 5 min at 500 × g on glass coverslips. Fixation, permeabilization and blocking were performed as previously described [59]. Cells were then stained with an anti-microtubule associated protein 1 light chain 3 beta (LC3B) rabbit polyclonal antibody (Sigma-Aldrich; L7543) and goat anti-rabbit IgG Alexa-Fluor 594 secondary antibody (ThermoFischer Scientific). Nuclei were counterstained with Hoechst 33342 (Invitrogen). GFP fluorescence from infected cells was directly observed. Cells were observed under an Axiovert 200 M inverted epifluorescence microscope equipped with a 40× PlanNeofluar oil/Dic objective or a 63× PlanApochromat oil/DIC and the Apotome imaging system (Zeiss, Jena, Germany). Images were captured with a CCD Axiocam MRm camera (Zeiss) by using the Axiovision software. Staining were done in triplicate and a minimum of 50 cells were

analyzed for each replicate in order to count the number of Lc3-positive cells and thus evaluate the proportion of cells in autophagy.

## Statistical analysis

All graphs and statistical analysis were performed using the GraphPad Prism software version 5.02 (San Diego, USA). Data are presented as means and standard deviations (±SD) or medians. One-way ANOVA was used to compare differences in multiple groups and a Mann-Whitney (two-tailed) test was used to compare nonparametric variables between two groups. $P$ values <0.05 were considered statistically significant as indicated in the figure legends.

## Supporting information

**S1 Fig. Chemokines are differentially expressed in MDV-infected B-cells.** Microarray analysis revealed that 18 chemokines/chemokines receptors genes involved in cytokine/cytokine receptors interaction pathways are differentially expressed in RB1B-GFP-infected B-cells. Results are presented as fold change (FC) for the mRNA expression of each gene in MDV-infected B-cells relative to uninfected B-cells.
(TIF)

**S1 Table. Biological processes and signaling pathways affected upon MDV infection in B-cells.**
(XLSX)

**S2 Table. Differentially expressed cytokines/cytokine receptors in infected B-cells.**
(DOCX)

**S3 Table. Differentially expressed genes involved in cell cycle pathways.**
(DOCX)

**S4 Table. Differentially expressed genes involved in apoptosis.**
(DOCX)

**S5 Table. Differentially expressed genes involved in autophagy.**
(DOCX)

**S6 Table. Differentially expressed genes involved in senescence.**
(DOCX)

## Acknowledgments

The authors would like to thank Yves Le Vern (INRAE, Nouzilly, France) for FACS analysis and Dr. Venugopal Nair (The Pirbright Institute, UK) for providing the recombinant RB-1B_pTK-GFP virus and Camille Berthault for technical assistance.

## Author Contributions

**Conceptualization:** Laëtitia Trapp-Fragnet, Benedikt B. Kaufer, Sonja Härtle.

**Data curation:** Laëtitia Trapp-Fragnet, Sonja Härtle.

**Formal analysis:** Laëtitia Trapp-Fragnet, Florian Pfaff, Sonja Härtle.

**Funding acquisition:** Caroline Denesvre, Benedikt B. Kaufer, Sonja Härtle.

**Investigation:** Laëtitia Trapp-Fragnet, Julia Schermuly, Marina Kohn, Luca D. Bertzbach, Benedikt B. Kaufer, Sonja Härtle.

**Methodology:** Laëtitia Trapp-Fragnet, Luca D. Bertzbach, Benedikt B. Kaufer, Sonja Härtle.

**Project administration:** Laëtitia Trapp-Fragnet, Sonja Härtle.

**Resources:** Laëtitia Trapp-Fragnet, Sonja Härtle.

**Supervision:** Laëtitia Trapp-Fragnet, Sonja Härtle.

**Validation:** Laëtitia Trapp-Fragnet, Sonja Härtle.

**Visualization:** Laëtitia Trapp-Fragnet, Sonja Härtle.

**Writing – original draft:** Laëtitia Trapp-Fragnet, Luca D. Bertzbach, Benedikt B. Kaufer, Sonja Härtle.

**Writing – review & editing:** Laëtitia Trapp-Fragnet, Luca D. Bertzbach, Caroline Denesvre, Benedikt B. Kaufer, Sonja Härtle.

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
