## [Decision Letter · Decision Letter 0]

20 Apr 2021

Dear Dr. TRAPP-FRAGNET,

Thank you very much for submitting your manuscript "Marek’s disease virus prolongs cell survival of primary chicken B-cells by inducing a senescence-like phenotype." for consideration at PLOS Pathogens. As with all papers reviewed by the journal, your manuscript was reviewed by members of the editorial board and by several independent reviewers. In light of the reviews (below this email), we would like to invite the resubmission of a significantly-revised version that takes into account the reviewers' comments.

We cannot make any decision about publication until we have seen the revised manuscript and your response to the reviewers' comments. Your revised manuscript is also likely to be sent to reviewers for further evaluation.

Sincerely,

Zhen Lin

Associate Editor

PLOS Pathogens

Erik Flemington

Section Editor

PLOS Pathogens

Kasturi Haldar

Editor-in-Chief

PLOS Pathogens

orcid.org/0000-0001-5065-158X

Michael Malim

Editor-in-Chief

PLOS Pathogens

orcid.org/0000-0002-7699-2064

Reviewer's Responses to Questions

**Part I - Summary**

Reviewer #1: In this report by Trapp-Fragnet and colleagues, they use their established in vitro Marek’s disease virus (MDV) infection model to study B cell infection. Although not perfect and artificial, this method allows them to study replication in primary B cells and perform many downstream events. Here they examine how MDV affects B cell survival, proliferation, autophagy, and senescence, among other things. First they found that MDV-infected B cells survived longer than uninfected B cells in culture. Through an exhaustive analysis, they systematically examined multiple pathways that could explain the survivability of MDV-infected B cells and found senescence was key to this phenomenon. Contrary to original thinking that MDV induced atrophy in lymphoid organs, such as the bursa, was due to apoptosis in B cells infected with MDV, they found that B cells infected with MDV were actually protected from apoptosis and survived longer than uninfected B cells. This was a well-written manuscript, but a few issues should be addressed.

Reviewer #2: (No Response)

Reviewer #3: In this manuscript, Trapp-Fragnet et al. explore the impact of Merek’s Disease Virus (MDV) on cell survival in vitro. They utilize a co-culture system, previously developed by the lab, to infect primary chicken bursal cells with a particularly productive MDV strain. Having already established an in vitro model of MDV infection, in this manuscript, the authors characterize the phenotypic impact of MDV on host cells during the initial stages of infection. The authors use microarrays, followed by pathway analysis, to compare gene expression between infected and non-infected cells. They find that MDV-infected B cells outlive non-infected controls in vitro, demonstrating that while fewer infected cells proliferate, they are also significantly less apoptotic than their non-infected controls.

**Part II – Major Issues: Key Experiments Required for Acceptance**

Reviewer #1: 1) The first experiment suggesting B cell viability is confusing. The authors use the term “control” for multiple groups in the results, materials and methods, and figure legend that makes it confusing as to what they are comparing. For example, the text (line 104) says bursal cells treated with CD40L but not with CEC is a control. Control for what and is this the control in Figure 1? If so, this is not the proper control for this experiment. I surmised the “control” in Figure 1 are viable cells collected by FACs that were not GFP positive, as this would be a proper control, but it is not clear in the text..

2) Measuring the total number of cells following incubation with uninfected and infected cells is flawed (Fig. 1). It is not clear if the viable non-GFP and GFP cells obtained by FACS started at the same number of cells. It is mentioned for RNA samples, 3 million cells were used, but it is not clear if this same sample preparation was used for Figure 1. Showing the total number of viable cells at 48 and 72 hours is not justified as the total number of viable cells at days 0 (after incubation on CECs) and 24 hours should also be shown. Unless the total number of cells at day 0 was same for both groups. It would be more logical to represent the data as the % viable over time (if the numbers were different) since the number of viable cells is most likely different from the start. This should be either shown in the data or represented better.

3) The data on autophagy is not convincing (Fig 6; S2). To verify the L3B antibody works on chicken cells and is a marker for autophagy, the authors need to use a chemical inducer of autophagy as a control. Likewise, EM of potential autophagosomes is not convincing as the image shown is not clear. Even when zooming in, it is difficult to see the double-walled autophagosomes and using a stress inducing agent as a control would help in this regard.

Reviewer #2: Using cultivation of B cells, FACS sorting, and Microarrays, the authors show that infected B cells are in a state of senescence, which might allow the virus more time to replicate or more time to infect T-cells. All this work was done using their B cell model, so the contribution of the other cells in the infected Bursa was excluded but likely important in deciphering what actually is happening in birds. They eloquently connected their microarray data with FACS analysis to determine what biological pathways are most likely affected in the infected B cells. Other researchers usually show what genes go up and down without making any connection to a biological function, not this group! Frankly really liked this manuscript, although I would like to see more modern techniques being used (i.e., the single-cell transcriptome of cells in the bursa). An atlas of MDV infected Birsa, so the speak.

Reviewer #3: 1. The authors establish infection by culturing bursal cells with infected chicken embryo cells, then purifying infected B-cells via FACS. However, their uninfected control bursal cells were never incubated with chicken embryo cells, leaving open the possibility that the observed differences in survival, apoptosis, cell cycle progression, and gene expression were impacted, at least in part, by one or more factors secreted by the chicken embryo cells. To account for this possibility, a more suitable control may be to co-culture uninfected chicken embryo cells with bursal cells.

2. Based on apoptosis and cell cycle analyses, the authors conclude that MDV-infected cells escape cell death by undergoing senescence. This hypothesis should be directly tested using B-gal staining or another established senescent cell detection system. It may be possible that MDV-infected cells delay apoptosis by a day or two without ever truly transitioning to long-term senescence.

3. This study lacks mechanistic detail and much of the data is descriptive. It is important to understand how viral infection increases host cell viability.

**Part III – Minor Issues: Editorial and Data Presentation Modifications**

Reviewer #1: 1) In Figure 5b, is the Meq-null virus derived from the control (RB1B-GFP) that has pUL47 tagged? Or is it derived from the TK-GFP virus? In other words, is it being compared to its parent virus?

2) Line 48; MDV is Gallid alphaherpesvirus 2.

3) Line 109; “contrast”

4) Line 306; “cytokines”

Reviewer #2: There are no major or so that, no minor issues, just the use of the word "precise" as a verb. This was odd. More commonly it is used as "to more precisely define."

Also using "for the first time" is unnecessary.

In this sentence, "Though on first glance a peculiar feature, the senescence-like phenotype of bursal

386 leukocytes could promote MDV life cycle and/or the persistence of the virus thus allowing it

387 to have more time to recruit and infect T-cells," delete "MDV life cycle and/or." The persistence of MDV is part of the life cycle.

Reviewer #3: 1. Autophagy results in Fig 6B should be more quantitative (maybe by counting cells with and without marker), and include statistics.

2. The conclusion stated in the subheader, “MDV infections strongly regulate gene expression in B-cells” (line 114), is

3. Microarray data in Figs 2A, 3A, 4A, 6A, and 7A should indicate significance and include error bars. Or apply rank-order statistical tests on all genes of a given pathway (i.e. KS test or similar). Somehow the variation should be taken into consideration.

4. Y-axis of microarray data should indicate that it is log(fold change) not just fold change. I assume log because there are negative values – y-axis should indicate the appropriate calculation.

PLOS authors have the option to publish the peer review history of their article (what does this mean?). If published, this will include your full peer review and any attached files.

Reviewer #1: No

Reviewer #2: No

Reviewer #3: No
---

## [Decision Letter · Decision Letter 1]

30 Jul 2021

Dear Dr. TRAPP-FRAGNET,

Thank you for submitting your manuscript "Marek’s disease virus prolongs survival of primary chicken B-cells by inducing a senescence-like phenotype." for consideration at PLOS Pathogens. Your revised manuscripts were re-evaluated by the original reviewers.  Although the reviewers agreed that the overall quality of the revised manuscript was significantly improved, a serious concern regarding the lack of rigor of the microarray experiment remains unaddressed.  We highly recommend that at a minimum, the appropriate control be included for the microarray analysis prior to submission of a revised version.

We cannot make any decision about publication until we have seen the revised manuscript and your response to the reviewers' comments. Your revised manuscript is also likely to be sent to reviewers for further evaluation.

Sincerely,

Zhen Lin

Associate Editor

PLOS Pathogens

Erik Flemington

Section Editor

PLOS Pathogens

Kasturi Haldar

Editor-in-Chief

PLOS Pathogens

orcid.org/0000-0001-5065-158X

Michael Malim

Editor-in-Chief

PLOS Pathogens

orcid.org/0000-0002-7699-2064

Your revised manuscripts were re-evaluated by the original reviewers. Although the reviewers agreed that the overall quality of the revised manuscript was significantly improved, a serious concern regarding the lack of rigor of the microarray experiment remains unaddressed. I suggest the author to include the appropriate control in the microarray analysis.

Reviewer's Responses to Questions

**Part I - Summary**

Reviewer #1: This report is highly improved and I am satisfied with the authors' revisions.

Reviewer #3: My original concerns regarding the lack of a proper control were only partially addressed. The authors repeated their cell cycle and proliferation experiments, this time including what I believe to be the more definitive control, comparing:

a) infected B cells + chicken embryo cells

b) uninfected B cells + chicken embryo cells

In their response, they show that the inclusion of chicken embryo cells in group b has little effect on BRDU incorporation and cell cycle stage, and thus this control is unnecessary for all other experiments conducted in this study. For the rest of the manuscript (including global gene expression analysis), they compare:

a) infected B cells + chicken embryo cells

b) uninfected B cells

Comparing these two groups, they conclude that “MDV infection protects B-cells from apoptosis”, and “Autophagy is induced in MDV-infected B-cells”, and they suggest this occurs as a result of, or as part of, an MDV-driven senescence program. It is unclear how much of a role the chicken embryo cells play in these processes.

The authors were unable to conclusively demonstrate senescence due to difficulties establishing an SA B-gal protocol for avian cells.

The added percentages of LC3-positive cells look very nice and provide great support.

**Part II – Major Issues: Key Experiments Required for Acceptance**

Reviewer #1: None

Reviewer #3: (No Response)

**Part III – Minor Issues: Editorial and Data Presentation Modifications**

Reviewer #1: None

Reviewer #3: (No Response)

PLOS authors have the option to publish the peer review history of their article (what does this mean?). If published, this will include your full peer review and any attached files.

Reviewer #1: **Yes: **Keith W. Jarosinski

Reviewer #3: No
---

## [Decision Letter · Decision Letter 2]

4 Oct 2021

Dear Dr. TRAPP-FRAGNET,

We are pleased to inform you that your manuscript 'Marek’s disease virus prolongs survival of primary chicken B-cells by inducing a senescence-like phenotype.' has been provisionally accepted for publication in PLOS Pathogens.

Best regards,

Zhen Lin

Associate Editor

PLOS Pathogens

Erik Flemington

Section Editor

PLOS Pathogens

Kasturi Haldar

Editor-in-Chief

PLOS Pathogens

orcid.org/0000-0001-5065-158X

Michael Malim

Editor-in-Chief

PLOS Pathogens

orcid.org/0000-0002-7699-2064

Reviewer Comments (if any, and for reference):

Reviewer's Responses to Questions

**Part I - Summary**

Reviewer #3: (No Response)

**Part II – Major Issues: Key Experiments Required for Acceptance**

Reviewer #3: (No Response)

**Part III – Minor Issues: Editorial and Data Presentation Modifications**

Reviewer #3: (No Response)

PLOS authors have the option to publish the peer review history of their article (what does this mean?). If published, this will include your full peer review and any attached files.

Reviewer #3: No

---

## [Editor Report · Acceptance letter]

15 Oct 2021

Dear Dr. Trapp-Fragnet,

We are delighted to inform you that your manuscript, "Marek’s disease virus prolongs survival of primary chicken B-cells by inducing a senescence-like phenotype.," has been formally accepted for publication in PLOS Pathogens.

Best regards,

Kasturi Haldar

Editor-in-Chief

PLOS Pathogens

orcid.org/0000-0001-5065-158X

Michael Malim

Editor-in-Chief

PLOS Pathogens

orcid.org/0000-0002-7699-2064